# `strategic-fl-sim`: An Extensible Package for Simulating Strategic Behavior in Federated Learning

**Dimitar Chakarov**
Toyota Technological Institute at Chicago
chakarov@ttic.edu

**Nikola Konstantinov**
INSAIT, Sofia University "St. Kliment Ohridski"
nikola.konstantinov@insait.ai

## Abstract

We introduce `strategic-fl-sim`, a lightweight research package for *strategic behavior* in Federated Learning (FL). Most existing FL frameworks target deployment or other simulation settings—such as privacy, benchmarking or heterogeneity—and are not designed for modeling strategic clients. `strategic-fl-sim` fills this gap by prioritizing easy specification of **client strategies** (transformations of local updates) and **server defenses** (aggregation methods). The design enforces a clean client–server separation: the `Client` owns data, local optimizers, local training, and strategic actions, while the `Server` handles aggregation, global updates and records metrics. Out-of-the-box, `strategic-fl-sim` includes implementations of common manipulations and aggregators, while remaining extensible. It supports single-node multi-GPU execution for efficient simulations in heterogeneous settings. We showcase the package's utility for the FedAvg protocol and three strategic actions—gradient scaling, sign-flipping, and free-riding. The project repository is available at `https://github.com/dimitarch/strategic-fl-sim`.

## 1   Introduction

Federated learning (FL) enables collaborative model training without centralized data [20, 16]. The distributed setup creates opportunities for *strategic client behavior*. Clients may manipulate updates to gain advantage [8, 4], free-ride to reduce computation [17], or launch adversarial attacks against the global model [30]. Understanding these dynamics is crucial for building robust FL protocols and effective incentive mechanisms. Existing FL frameworks, however, are not designed for this purpose. Production-oriented packages such as Flower [1], PySyft [32], and FEDn [9] emphasize deployment priorities like communication, privacy, and scalability. Research-oriented frameworks such as FedML [14], FedScale [19], pfl-research [12], and Pollen [24] focus on heterogeneity, benchmarking, or privacy, and FL-Byzantine [23, 22] and byzfl [11] are geared towards Byzantine-robustness. While these existing frameworks serve their goals effectively, they do not explicitly focus on strategic behavior. As a result, researchers of strategic FL interactions often rely on manual client coordination, which makes experiments both cumbersome and difficult to replicate.

### 1.1   Our Contribution

`strategic-fl-sim` addresses this gap with a general-purpose, extensible toolkit for modeling strategic behavior in FL. It prioritizes convenient simulation over deployment speed and is therefore particularly suited for research on incentives in FL and robustness dynamics. Our main contributions are:

- We introduce `strategic-fl-sim`, the first FL simulation package designed specifically for strategic behavior simulation.

39th Conference on Neural Information Processing Systems (NeurIPS 2025) Workshop on Reliable ML from Unreliable Data.

- We provide clean abstractions separating client-side strategies (arbitrary transformations of updates) from server-side logic (aggregation methods, metrics), with extensible implementations of both.
- We enable single-node, multi-GPU execution out-of-the-box, supporting high-throughput simulations.
- We demonstrate the package on three datasets—FeMNIST, Shakespeare, and Twitter/Sent140 [3]—and three strategic actions: gradient scaling, free-riding, and sign-flipping.

We aim for `strategic-fl-sim` to support the research community in systematically studying strategic client behavior and server defenses, facilitating reproducible streamlined experimentation and robust evaluation of FL protocols.

### 1.2 Related work

`strategic-fl-sim` complements existing FL packages by addressing a specific research need unmet by current tools. We organize related work into four categories: production packages, research simulation packages, specialized packages and prior work on incentives in collaborative learning.

**Production-oriented packages.** Flower [1], PySyft [32], and FEDn [9] excel at real-world deployment concerns including network communication, security protocols, and distributed fault tolerance. However, these production priorities create barriers for strategic behavior research: setting up heterogeneous client populations with different strategic actions requires extensive customization and careful coordination of client states across distributed systems. Research packages like FedML [14] focus on system heterogeneity simulation rather than strategic dynamics.

**Research simulation packages.** Several packages target FL research simulation but with different emphases than `strategic-fl-sim`. FedScale [19] provides comprehensive benchmarking with realistic datasets and scalable runtime, focusing on system heterogeneity and performance evaluation. pfl-research [12] offers high-performance simulation with strong privacy algorithm integration, achieving 7-72× speedup over alternatives on cross-device setups. Pollen [24] addresses scalability through resource-aware client placement and high-throughput simulation. While these packages excel at their respective goals—benchmarking, privacy-preserving FL, and scalable simulation—they do not provide the specialized abstractions needed for strategic behavior research.

**Specialized research packages.** Most related are Byzantine-robustness packages like FL-Byzantine [23, 22] and byzfl [11], which implement SOTA attacks and defenses for Byzantine FL. Our focus differs by providing a tool for implementing arbitrary client and server behaviors. Our package abstracts client logic (data, optimizer, local training, strategic actions) and server logic (aggregation, global training, coordination, metrics logging) , prioritizing extensibility through modularity. This allows researchers to implement any server and client behavior, including methods beyond existing SOTA Byzantine approaches. Other recent works like APFL [10] and PPFL [21] address specific strategic FL aspects through algorithmic contributions for non-IID challenges and heterogeneous population patterns. However, these lack general-purpose tools for strategic behavior simulation.

**Incentives in Collaborative Learning.** There have been several prominent directions of theoretical research that could benefit from our unified toolkit. For instance, prior work has investigated voluntary participation and individual rationality [28, 31], and explored the impact of privacy concerns on FL participation incentives [27]. Some of the works that consider non-truthful participation focus on free-riding [17, 26], study defection [13], look at agents who wish to keep their data collection low [2], incentivize diverse and high-quality data contributions [15, 29], or consider manipulating updates due to competition in homogeneous settings [8]. In heterogeneous settings, earlier works have successfully designed truthful mechanisms for single-round mean estimation [5, 7] and peer prediction [6], and federated stochastic gradient descent under strategic positive definite transformations [4].

## 2 Theoretical framework

We consider a standard FL setting with $N$ clients collaborating through a central server. All clients work with a shared loss function $f(\theta; z)$ where $\theta \in \mathbb{R}^d$ and $z \in \mathcal{Z}$. Each client $i$ has their own data distribution $D_i$ over $z \in \mathcal{Z}$, and their objective is $F_i(\theta) = \mathbb{E}_{z \sim D_i}[f(\theta; z)]$. The server's objective is to minimize the average expected loss $F(\theta) = \frac{1}{N} \sum_{i=1}^{N} F_i(\theta)$.

**Strategic Client Actions.** The client action space $\mathcal{A}$ consists of gradient transformations $\mathbf{a} : \mathbb{R}^d \to \mathbb{R}^d$ that clients can apply to their local updates. We may assume that $\mathcal{A}$ always contains the identity action $\mathbf{id}(g) = g$, representing honest behavior. For instance, the action space might contain gradient scaling, free-riding or sign flipping. At round $t$, client $i$ chooses an action $\mathbf{a}_t^i \in \mathcal{A}$, computes a gradient update $g_t^i(\theta_t)$, which may be a simple stochastic gradient, the result of several local SGD steps or any other local training method, and sends the transformed message $m_t^i = \mathbf{a}_t^i(g_t^i(\theta_t))$ to the server. The server then aggregates these potentially manipulated gradients and updates the global model as described below (Section 2).

**Server Protocol.** We consider a general FL protocol $\mathcal{M}$ of the following type. At time $t$ client $i$ sends a gradient message $m_t^i \in \mathbb{R}^d$ to the server. The server aggregates all messages according to a fixed aggregation function $\mathtt{Aggregate} : \mathbb{R}^{d \times N} \to \mathbb{R}^d$, such as mean, median or trimmed mean aggregation, so $\bar{m}_t = \mathtt{Aggregate}\left(\{m_t^i\}_{i=1}^N\right)$. Then it updates the central model $\theta_{t+1} = \theta_t - \gamma_t \bar{m}_t$, where $\gamma_t$ is the learning rate at step $t$. Finally, the server broadcasts $\theta_{t+1}$ to all clients. This process is repeated for $T$ steps. Furthermore, in the strategic FL literature often the server implements defenses that calculate a gradient-based client payment [8, 4], complementing robust aggregation.

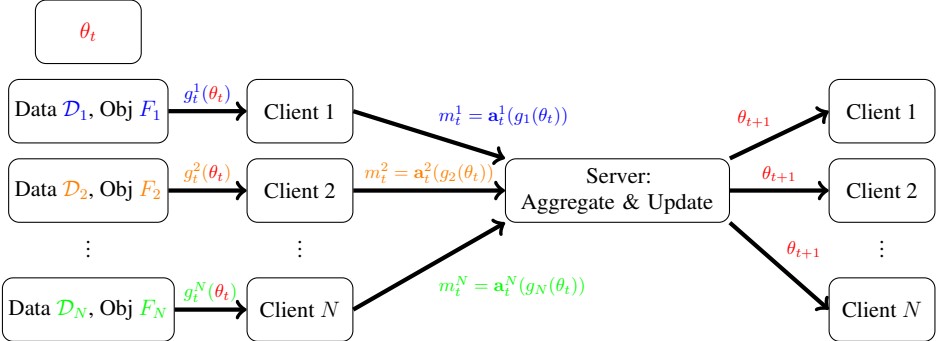

Figure 1: Example with strategic actions in an oversimplified FL protocol.

## 3 The `strategic-fl-sim` Package Architecture

`strategic-fl-sim` is implemented in PyTorch with minimal dependencies (NumPy, OmegaConf for configurations). The architecture directly translates the theoretical framework from Section 2 into a practical research tool. The project repository is available at `https://github.com/dimitarch/strategic-fl-sim`.

### 3.1 Design principles

`strategic-fl-sim`'s architecture reflects the following three core principles.

**Research-first philosophy.** `strategic-fl-sim` prioritizes ease of experimentation and simulation over deployment concerns, such as network protocols, security, or distributed fault tolerance.

**Clean Client-Server separation.** Clients and servers have clearly separated responsibilities. Clients own their data, local optimization, training procedures, and strategic actions. The server exclusively handles aggregation, global model updates, and system-wide metrics collection.

**Extensibility by default.** Both strategic actions and aggregation methods come with default functions, but are designed for easy customization. Adding new strategic behaviors or defense mechanisms requires minimal code changes, encouraging rapid prototyping and comparative studies.

### 3.2 Overview of architecture

The theoretical setup from Section 2 defines clients as entities that own data and apply strategic actions $\mathbf{a}_t^i$, while servers handle aggregation functions $\mathtt{Aggregate}(\cdot)$ and global model updates. The software architecture enforces this separation through two main classes that encapsulate distinct responsibilities and don't cross boundaries. This design choice serves a critical research purpose:

investigators can modify client strategies independently of server defenses, and vice versa, enabling controlled experiments where exactly one variable changes at a time.

Consider the fundamental research workflow in strategic federated learning: researchers typically want to compare how different aggregation methods perform against the same strategic population, or evaluate how a fixed aggregation method handles different types of strategic behavior.

`strategic-fl-sim`'s clean separation makes such studies straightforward—swap the aggregation function without touching client code, modify client actions without changing server logic, customize the server training methodology through subclassing the existing class with no modifications to the client logic and vice versa.

### 3.3    Core API example

The following example demonstrates how `strategic-fl-sim`'s API directly reflects the theoretical framework while remaining simple to use:

```python
# Step 1: Define what each client owns and how they behave
honest_client = Client(
    model=CNN(),                      # Local copy of generic
    PyTorch CNN model
    data_loader=honest_data,          # Private dataset
    action=create_scalar_action(1.0)  # Strategy: honest reporting
)

strategic_client = Client(
    model=CNN(),
    data_loader=strategic_data,       # Different data distribution
    action=create_scalar_action(2.0)  # Strategy: 2x gradient
    scaling
)

byzantine_client = Client(
    model=CNN(),
    data_loader=byzantine_data,
    action=create_sign_flip_action()  # Strategy: Byzantine attack
)

# Step 2: Define what the server controls globally
server = Server(
    model=global_model,               # Global model state
    theta_t
    aggregate_fn=get_aggregate("mean"), # Aggregation function
    Aggregate(.)
    metrics_hook=get_gradient_metrics   # Metrics hook for logging
)

# Step 3: Package coordinates interaction automatically
clients = [honest_client, strategic_client, byzantine_client]
losses, metrics = server.train(clients=clients, T=100)
```

This API enforces our theoretical framework in practice. Clients own their datasets, local training procedures, and strategic action functions $\mathbf{a}_t^i$. The server owns the global model state $\theta_t$, aggregation mechanism $\text{Aggregate}(\cdot)$, the metrics gathering function and system-wide coordination. Neither side needs implementation details of the other—the package handles all message passing, device coordination, and state synchronization automatically.

### 3.4    Core components

The clean theoretical separation translates into clear software responsibilities that enable different types of strategic behavior research.

`Client` **Class.** Local data loaders and model parameters enable heterogeneity studies by allowing different clients to have different data distributions $D_i$ and different local model initializations. Local training procedures and optimizers enable studies of how different local update methods $g_t^i(\theta_t)$ interact with strategic actions—for instance, comparing how gradient scaling affects single-step SGD versus multi-step local training. Strategic action functions $\mathbf{a}_t^i$ allow for transforming computed gradients before transmission to the server. Finally, device placement enables both high-throughput simulation and modeling of federated environments where clients have different computational capabilities. Clients perform only local computations and strategic transformations. They never access information from other clients, can't perform aggregation, and can't update the global model directly. This restriction ensures that strategic clients operate within realistic federated constraints.

`Server` **Class.** The server handles everything global in the federated learning protocol. It manages the global model state $\theta_t$ and applies aggregation mechanisms $\texttt{Aggregate}\left(\{m_t^i\}_{i=1}^N\right)$ to transform client messages into global updates. The server coordinates cross-device tensor transfers when clients are distributed across multiple GPUs. The server never accesses client private data, local optimizers, or strategic action implementations. This separation ensures that aggregation methods operate only on the information that would be available in real federated deployments.

**Metrics hooks.** The server also maintains a *metrics collection* through customizable metrics hooks that can compute gradient norms, cosine similarities, outlier detection scores, and other indicators of strategic behavior without storing complete gradient histories. This allows the server to also compute client payment schemes: for example, the payment schemes from Chakarov et al. [4] and Dorner et al. [8] can be implemented because all gradients can be tracked via the metrics hooks.

### 3.5 Extensibility

The separation between client and server responsibilities enables the modification of either side independently without breaking the other. Clients can implement arbitrarily sophisticated strategic behaviors: adaptive strategies using model history, collusion simulation, data-dependent actions. As long as the client produces a tensor of appropriate dimensions to send to the server, the package handles the rest automatically. Similarly, the server can implement complex robust mechanisms, adaptive defenses that change based on detected strategic behavior, or incentive-compatible mechanisms. The aggregation function receives the client messages $\{m_t^i\}_{i=1}^N$, ensuring that defensive mechanisms remain realistic. Below is an example of a custom strategic action based on gradient magnitude:

```python
def magnitude_adaptive_scaling(gradient, scaling_factor=2.0,
    threshold=0.1):
    grad_norm = torch.norm(gradient)
    if grad_norm < threshold:   # Scale aggressively when small
        return gradient * scaling_factor
    else:
        return gradient  # Honest behavior for large gradients

client = Client(model=CNN(), data_loader=data, action=
    magnitude_adaptive_scaling)
```

### 3.6 Single-node multi-GPU execution

`strategic-fl-sim`'s device abstraction serves dual purposes beyond simple performance improvement: (1) it enables high-throughput simulation by distributing clients across multiple GPUs on a single node, and (2) it supports modeling federated environments where clients have heterogeneous computational capabilities by placing clients on devices with different memory and processing constraints. The package handles device coordination automatically during aggregation and broadcasting, transferring tensors between client devices and the server device as needed.

```python
# Distribute clients across available GPUs
client1 = Client(model=CNN(), data_loader=data1, device="cuda:0")
client2 = Client(model=CNN(), data_loader=data2, device="cuda:1")
client2 = Client(model=CNN(), data_loader=data2, device="cuda:2")
server = Server(model=CNN(), aggregate_fn=median, device="cuda:3")
```

### 3.7 Built-in components for common research patterns

`strategic-fl-sim` ships with a set of common strategic actions and aggregation methods from the strategic FL literature. These components function as both immediately usable tools and templates that could be adapted for novel studies. For instance, a gradient scaling action can be extended to include adaptive scaling based on training round, local data characteristics, or observed server responses. Similarly, aggregation methods can incorporate learned parameters, multi-round memory, or incentive-compatible scoring mechanisms.

The built-in strategic actions cover the following. Gradient scaling [8, 4] attacks are implemented through `create_scalar_action(alpha, beta)`, which models upscaling attacks that amplify client influence ($\alpha > 1$), free-riding [18, 26] behaviors where clients minimize effort while benefiting from others ($\alpha < 1$), and noisy reporting scenarios with controlled variance injection (non-zero $\beta$). Byzantine attacks [30, 25] are supported through `create_sign_flip_action()` for coordinated malicious behavior and `create_coordinate_attack()` for dimension-specific perturbations.

Server-side defenses span from standard baselines to robust aggregation mechanisms [30]. Basic methods like arithmetic mean and weighted mean provide essential baselines for measuring attack effectiveness, while robust alternatives including coordinate-wise median, trimmed mean with configurable percentile cutoffs, and coordinate-wise trimmed mean offer graduated levels of Byzantine tolerance.

## 4 Experimental studies

We demonstrate `strategic-fl-sim`'s capability through three strategic behavior scenarios using the FeMNIST, Shakespeare and Twitter/Sent140 datasets from LEAF [3]. Each experiment isolates a specific strategic action—scaling, free-riding, sign-flipping. Each experiment requires only creating the necessary agents (clients and server) and calling the training function of the server.

**Gradient scaling on FeMNIST.** We use a two-layer CNN classifier: the first layer has 3 input channels and 32 output channels, and the second layer has 32 input channels and 64 output channels. Both layers apply a $5 \times 5$ kernel with stride 1 and padding 2. After each layer we add ReLU activation and $2 \times 2$ max pooling with stride 2. We train on FeMNIST with 10 clients using FedAvg with 3 local steps with batch size 32 for $T = 3500$ rounds. One client applies gradient scaling with $\alpha = 2.0$ while others remain honest. Figure 2 shows that the scaling client achieves better final loss compared to all but one of the honest participants.

**Free-Riding on Shakespeare.** We use a two-layer LSTM with embedding dimension of 8, 80 classes and 256 hidden units per layer. We train on Shakespeare with 1 local step and batch size 16 for $T = 1000$ rounds. One client free-rides and sends zero gradients at each step. Figure 3 shows that free-riding gives competitive performance for the strategic client with no local computation.

**Byzantine Attack on Twitter/Sent140.** We use a two-layer linear classifier with 384 hidden neurons on top frozen BERT embeddings. We train on Twitter with 5 clients using FedAvg with 1 local step and batch size 8 for $T = 500$ rounds. One client uses Byzantine sign-flipping. Figure 4 shows that this has minimal impact on convergence.

## 5 Limitations and future work

`strategic-fl-sim` currently targets single-node simulation, which is sufficient for strategic behavior research but limits scalability studies. The framework focuses on gradient-level strategic actions rather than model-level or data-level manipulations. Communication costs and network effects are not modeled, as these are secondary concerns for strategic behavior analysis. Future extensions could include: distributed multi-node execution for scalability research, communication cost modeling for practical deployment studies, and integration with mechanism design frameworks for incentive analysis.

**Acknowledgements.** DC acknowledges funding and compute resources from TTIC. DC would like to thank Adam Bohlander for several useful discussions about implementing and executing distributed training. This research was partially funded by the Ministry of Education and Science of Bulgaria (support for INSAIT, part of the Bulgarian National Roadmap for Research Infrastructure).

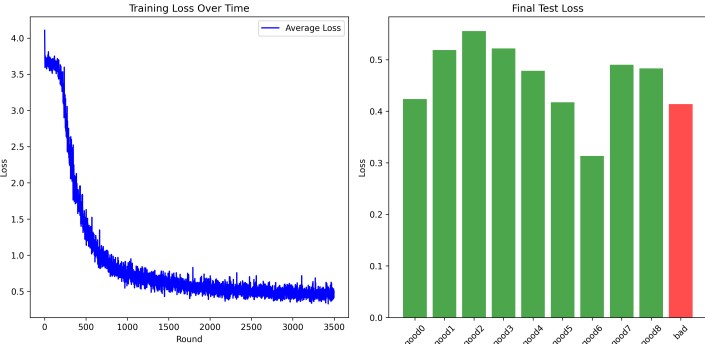

Figure 2: Training of CNN classifier with the FeMNIST dataset and 10 clients, where one is scaling their update by a factor of 2 and everyone else is honest. The protocol is FedAvg with 3 local steps and batch size 32 at each timestep, learning rate of $0.06$ and runs for $T = 3500$ steps.

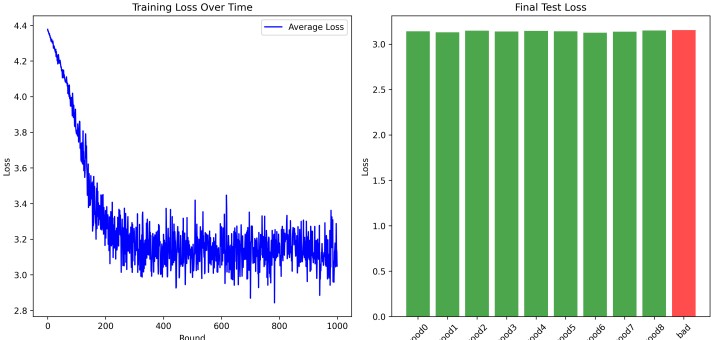

Figure 3: Training of LSTM with the Shakespeare dataset and 10 clients, where one is free-riding and everyone else is honest. The protocol is FedAvg with 1 local steps and batch size 16 at each timestep, mean aggregation, learning rate of $0.06$ and runs for $T = 1000$ steps.

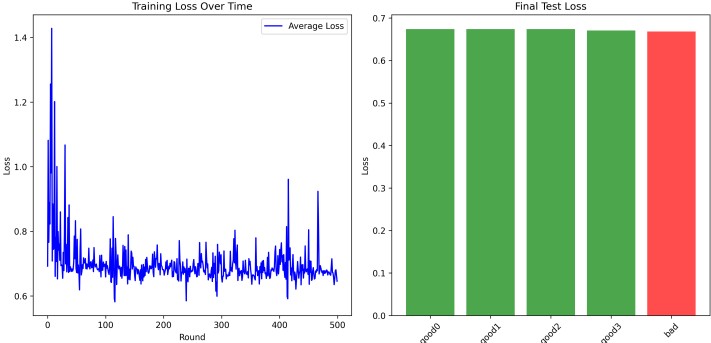

Figure 4: Training of BERT classifier with the Sent140/Twitter dataset and 5 clients, where one is using Byzentine sign flipping and everyone else is honest. The protocol is FedAvg with 1 local steps and batch size 8 at each timestep, mean aggregation, learning rate of $0.06$ and runs for $T = 500$ steps.

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
