# OpenReview forum: "$\texttt{strategic-fl-sim}$: An Extensible Package for Simulating Strategic Behavior in Federated Learning"
_NeurIPS.cc/2025/Workshop/Reliable_ML — NeurIPS 2025 - Reliable ML Workshop_

### Official Review · Reviewer_skQH · 2025-09-19
**Nice tool for research in robust federated learning**

**Rating:** 7
**Confidence:** 2

**Review:**

**Summary.** The paper introduces *strategic-fl-sim*, a new open-source Python package for simulating strategic client behaviors in Federated Learning. The authors claim that existing FL frameworks are primarily designed for deployment or other research areas like privacy and benchmarking, making them ill-suited for studying strategic actions. The proposed package fills this gap by providing a simple, extensible API built on a clean separation between client and server responsibilities.

The paper demonstrates the package's utility by simulating three common strategic actions on the FeMNIST, Shakespeare, and Sent140 datasets. The results show that the package can effectively model these scenarios and their impact on the learning process.

**Strengths.** Solid implementation of a very relevant research topic. The paper is also well-written and seems easy to use. The three experiments also effectively showcases the package's core functionalities.

**Weaknesses.** The paper is upright on its limitation that it only supports single-node simulation. Also, it'd be better if the paper demonstrates a more powerful Byzantine attack (right now it does little damage), and how the built-in defenses (e.g. trimmed mean) can help mitigate the damage.

**Suggestions.** It'd be nice to see the aforementioned Byzantine experiment. That would give a more complete and impressive demonstration.

---

### Official Review · Reviewer_2DYn · 2025-09-20
**Review for strategic-fl-sim**

**Rating:** 5
**Confidence:** 2

**Review:**

# Summary:
The paper introduces strategic-fl-sim, a lightweight and extensible research package designed to simulate strategic client behavior in federated learning. Unlike existing FL frameworks, which mainly focuses on deployment, privacy, benchmarking, or Byzantine robustness, this tool is explicitly built for modeling client strategic behaviors, making experimentation and reproducibility easier.

# Strengths:
1. The authors have done comprehensive literature review to discuss relevant packages in the field of federated learning
2. The authors provide a toolkit specifically targeting strategic behavior in FL, which seems to be a a gap not addressed by existing frameworks
3. The package seems to have several strengths, such as extensibility, lightweight, and easy to use.

# Weaknesses:
1. Currently limited to single-node execution; cannot model realistic multi-node distributed systems at scale.
2. The evaluation seems to be limited. The paper didn’t seem to demonstrate how much more effort it would take to run these experiments without the package

# Suggestions:
1. Would be very helpful if when the authors demonstrate three case studies, they can run head-to-head comparisons against “doing it without the package” (i.e., manual scripting or other frameworks) in terms of setup effort, runtime, or experimental reproducibility. Doing something like a quantitative “Ease-of-Use” study might make sense: comparing how much less code the users need to write, how much less time it takes for the experiment to be set up correctly, etc…
2. In introduction, it would be great to add some quick explanation to key terms such as “(clients may) free-ride to reduce computation”